# Performance Investigation of Superplastic Shape Memory Alloy-Based Vibration Isolator for X-Band Active Small SAR Satellite of S-STEP under Acoustic and Random Vibration Environments

**Hyun-Guk Kim** [1], **Seong-Cheol Kwon** [1], **Kyung-Rae Koo** [1], **Sung-Chan Song** [1], **Youngjoon Yu** [1], **Youngbum Song** [1], **Yeon-Hyeok Park** [2] and **Hyun-Ung Oh** [3,*]

1   Yongin R&D Center, Hanwha Systems, 491-23, Gyeonggidong-ro, Namsa-myeon, Cheoin-gu, Yongin-si 17121, Gyeonggi-do, Korea
2   Steplab, 66 Daehwa-ro 106beon-gil, Daedeok-gu, Daejeon 34365, Korea
3   Space Technology Synthesis Laboratory, Department of Smart Vehicle System Engineering, Chosun University (Agency for Defense Development: Additional Post), Pilmundae-ro, Dong-gu, Gwangju 61452, Korea
*   Correspondence: ohu129@chosun.ac.kr

**Abstract:** In a launch environment, all satellites are subjected to severe random vibration and acoustic loads owing to rocket separation, airflow, and injection/combustion of the fuel. Structural vibrations induced by mechanical loads cause the malfunction of vibration-sensitive components in a satellite, leading to failures during the launch process or an on-orbit mission. Therefore, in this study, a shape memory alloy-based vibration isolator was used on the connection between the launch vehicle and satellite to reduce the vibration transmission to a satellite. The vibration isolator exhibited a high performance in the vibration isolation, owing to the dynamic properties of super-elasticity and high damping. The vibration-reduction performance of the vibration isolator was experimentally verified using random vibration and acoustic tests in a structural thermal model of the satellite developed in the synthetic aperture radar technology experimental project. Owing to the super-elasticity and high attenuation characteristics of the vibration isolator, it was possible to significantly reduce the random vibration of the satellite in the launch environment. Although the mechanical load of the acoustic test mainly excited the antenna on the upper side of the satellite rather than the bottom side, the results of the acoustic test showed the same trend as the random vibration test. From this perspective, the vibration isolator can contribute to saving the costs required for satellite development. These advantages have made it possible to develop satellites according to the new space paradigm, which is a trend in the space industry worldwide.

**Keywords:** synthetic aperture radar (SAR); small SAR technology experimental project (S-STEP); random vibration test; acoustic test; shape memory alloy (SMA); structural thermal model (STM)

## 1. Introduction

The new space paradigm is changing the development philosophy of space engineering worldwide. The new space paradigm refers to the global gradual commercialization of the space engineering field driven by private companies rather than government organizations [1,2]. This acceleration has been enforced by the emergence of space philosophies toward faster, cheaper, and lighter spacecraft development. In particular, the emergence of small-satellite constellations can be considered one of the main elements of the new space paradigm. Compared with existing mid to large satellites, small satellite constellations have a fast design evaluation. In addition, recent rapid developments in manufacturing technology have made it possible to mass produce satellites. As a result of these advantages,

small satellite platforms are intensively used for various challenging tasks, such as real-time remote sensing, fast telecommunication, and the global internet [3–5].

During the launch process, all satellites are subjected to various dynamic loads [6,7] caused by static acceleration owing to engine thrust, sine vibration due to engine shutoff, and self-excited vibration due to incomplete fuel combustion. In addition, thrust noises cause random vibration, and the separation process between the launch pad and spacecraft generates a mechanical shock. In addition, a diffused acoustic field of approximately 130–140 dB is generated in the satellite fairing owing to fuel combustion noise and aerodynamic noise generated by the friction between the launching vehicle and air [8–11]. This diffuse acoustic field is one of the primary sources of satellite vibration. Considering this, dynamic load is serious and complicated as it can cause launching and satellite mission failures and mechanical/electrical malfunctions. Therefore, to improve the structural stability of satellites, it is essential to achieve a mechanical design that guarantees structural safety against vibration and acoustic noise in the launch environment.

There are two approaches to improving the structural stability of mechanical systems subjected to dynamic loads: active and passive control methods. Active vibration control is implemented with a combination of a sensor that analyzes vibration signals transferred to the mechanical system and an actuator that generates anti-phase vibrational signals [12,13]. The artificially generated vibrational signals in the active control system have the advantage of a high vibration reduction effect; however, the system power budget must be considered for additional electric devices such as sensors and actuators. However, passive vibration control has the advantage of not requiring additional power consumption. However, the vibration reduction performance is relatively low because only the structure/material of the mechanical system is changed. In addition, the active vibration control needs the additional sensor and actuator systems. Considering the characteristics of the space environment, which is very difficult to maintain and has a limited power budget for the satellite system, passive vibration control is more suitable for reducing the vibration of the satellite system. There is no additional mass or volume of the satellite system because this passive vibration control does not require an additional printed circuit board (PCB) or power supply unit (PSU) [14–16]. Additionally, it is expected that the notching process, often used in environmental testing, can be simplified. The advantages of this passive vibration-control-based vibration isolation system are in line with the new space paradigm for developing small satellites at an effective cost.

In this study, the vibration reduction performance of a shape memory alloy (SMA)-based high-damping vibration isolator was experimentally verified for the structural-thermal model (STM) of a small synthetic aperture radar (SAR) satellite using acoustic and random vibration tests. The SMA-based high-damping vibration isolator used in this study had two key characteristics: the super-elasticity of SMAs and the high damping properties of laminated structures [16]. Super-elasticity is an intrinsic property of SMAs caused by a stress-induced phase transformation generated in a phase transition state. Because of these characteristics, the structure did not undergo plastic deformation and recovered its original shape under unloading conditions. Based on these dynamic behavior characteristics, the SMA was first applied to reduce the vibration caused by pyro-shock impact [17] and was constructed as a multilayered structure using viscoelastic tape to achieve a high damping performance.

Kwon et al. [18] proposed an SMA-based vibration isolation structure to reduce random vibrations in the launch environment of satellites. In 2017, an SMA-based blade-type vibration isolator was applied to reduce the vibration of the cooler, which had a high vibration reduction performance compared with that of the conventional structure consisting of titanium. In 2019, a super-elastic SMA wheel was used to reduce the micro-jitter vibration of a two-axis gimbal-type X-band antenna [19]. Park et al. [20] designed a high-damping PCB board using a laminated structure of viscoelastic acrylic tape. This PCB board concept improved the fatigue life of the vibration-sensitive electronic units in satellites. In 2021, Kwon et al. [4] proposed an SMA-based vibration isolator to

improve the structural safety of a small SAR satellite subjected to random vibrations. This SMA-based vibration isolator achieved a vibration energy attenuation of approximately 85% based on the center of mass of a small SAR satellite.

In this study, it was experimentally verified that the vibration reduction performance of an SMA-based vibration isolator reduced the structural vibration caused by the diffused sound field inside the fairing. Acoustic and random vibration tests were performed using the STM for an SAR technology experimental project (S-STEP) satellite. Based on the test results, the vibration reduction performance of the SMA-based vibration isolator was validated for the acoustic and random vibration tests. In addition, because the random vibration test results generally envelope the acoustic test, the entire process of the environment test can be simplified, and the development cost for the satellite can be reduced.

Section 2 presents a brief description of the S-STEP, a research project for small SAR satellites. Section 3 presents the concept of an SMA-based vibration isolator and its vibration reduction performance. Section 4 introduces the experimental setup of the random vibration and acoustic tests, and shows the experimental results.

## 2. Brief Description of an S-STEP System

The S-STEP was a research project that established a smaller, lighter, faster, and compact process for designing, evaluating, and manufacturing small SAR satellites according to the new space paradigm. The small SAR satellite developed in this project was called the S-STEP satellite and was used for missions to obtain high-resolution images for purposes such as environmental investigation, surface mapping, and disaster monitoring [3–5].

Figure 1 shows the main missions and operational methods of the S-STEP satellite. The S-STEP satellite had several operation modes depending on the mission, such as ScanSAR, VideoSAR, and stripmap modes, which were the main missions [21,22]. The ScanSAR and VideoSAR modes had resolutions of 1 and 4 m, respectively. These modes tracked various conditions for different targets of interest. The high-resolution stripmap mode provided a resolution of 1 m and wide area monitoring. Images and videos acquired through the SAR antenna were stored in an integrated avionics unit (IAU) through the Wizard Link and were transmitted to the ground through X-band antenna (at a speed of 1 Gbps). In addition, communication of the satellite constellation via an S-band inter-satellite link (ISL) improved mission performance at the system level. The main performances of the S-STEP satellite are listed in Table 1 [3]. As seen in Figure 1b,c, the S-STEP satellite was a compact plate-type structure, which had the weight of 80 kg class, and the resolution range, acquisition time, and transmitted peak power were 1 m, 5 km, 60 s, and 2560 W, respectively. From this perspective, the S-STEP satellite had a very high performance-to-mass ratio.

**Table 1.** System specification of the S-STEP satellite [3].

| Specification | | Value |
|---|---|---|
| Mission lifetime | | 3 years |
| Mass | | 80.3 kg |
| Satellite size | | $1970 \times 1060 \times 200$ mm |
| Power | Generation | 340 W |
| | Save | 648 Wh |
| Inter-satellite link | | RF (X-band) |
| TMTC/image download link | | S-band/X-band |
| Pointing accuracy | | 0.085° |
| Resolution (25°) | | 1 m (Stripmap) |
| | | 4 m (ScanSAR) |
| | | 1 m (VideoSAR) |
| Swath (elevation $\times$ azimuth) | | $5 \times 420$ km (Stripmap) |
| | | $15 \times 420$ km (ScanSAR) |
| | | $5 \times 5$ km (VideoSAR) |
| Image acquisition time | | 60 s (Stripmap) |
| | | 10 s (VideoSAR) |

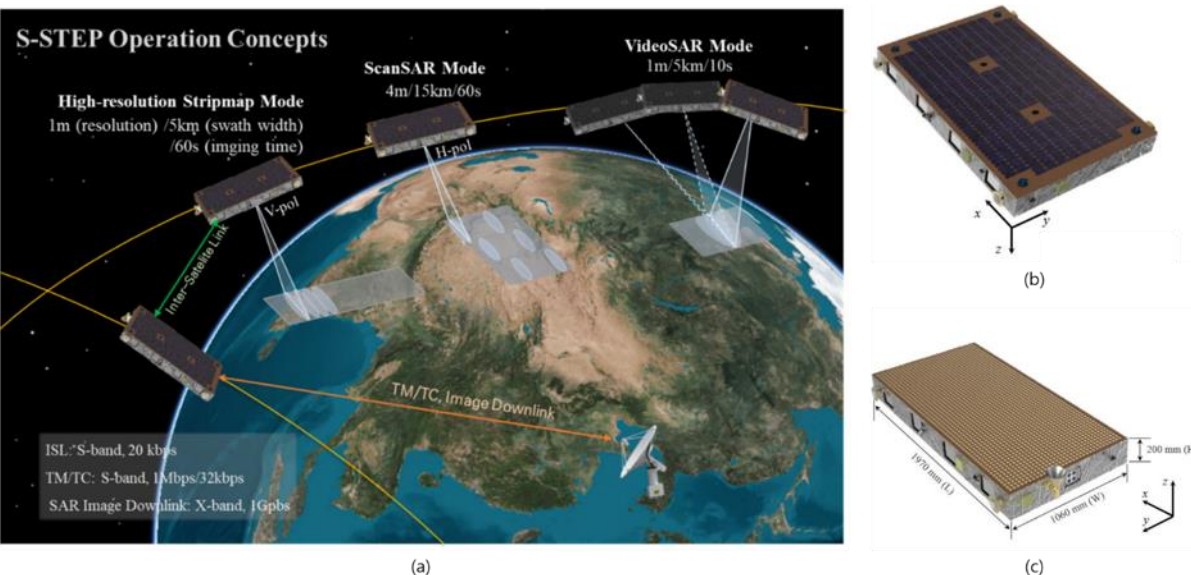

**Figure 1.** Schematic of S-STEP satellite: (**a**) S-STEP operation concepts of the mission; (**b**) bottom side with solar cell; (**c**) top side with SAR antenna [3].

Similar to most satellites, the environment tests of S-STEP satellites were implemented to verify the function and performance in the launch/orbit environment during the development process. In particular, the dynamic behavior of the S-STEP satellite caused by the random vibration (generated in the launch environment) and the acoustic load (from the interior acoustic field inside the fairing) should be investigated. This is because the vibro-acoustic effect in combination with the acoustic load and random vibration caused the malfunction of electronic components in S-STEP satellites. Consequently, mission failure of the S-STEP satellite could occur. Section 3 describes an SMA-based vibration isolator to minimize the vibration of the satellite in response to acoustic/vibration load acting in the launch environment.

## 3. Dynamic Behavior of SMA-Based Vibration Isolator

Figure 2 shows an SMA-based vibration isolator. The SMA-based vibration isolator used in this study consisted of two types of vibration-isolating elements for axial and lateral vibrations. As shown in Figure 2a, the vibration of the launch vehicle transferred from the outer bracket was transmitted to the inner bracket through the vibration-isolating element in the axial direction (Figure 2c), which was transferred to the upper plate through the vibration-isolating element in the lateral direction (Figure 2d). Vibration generated by the projectile is transmitted to the satellite through a vibration isolator. Here, the SMA-based vibration isolator reduced vibrations above the target cutoff frequency (28 Hz) with a high damping performance. This phenomenon is called "frequency decoupling", which means that the peak vibration value over the cutoff frequency can theoretically be reduced by up to 50%. This dynamic behavior of vibration isolation is caused by the super-elasticity of SMA-blade and the high damping property of viscoelastic tape.

This meant that the SMA-based vibration isolator could reduce the vibrations caused by sound/vibration excitation in the target frequency band. As shown in Figure 2e, it had high damping characteristics owing to the super-elastic properties of the vibration-insulating element, composed of a laminated structure consisting of an SMA blade, FR-4 material, and a constrained layer. Therefore, the SMA-based vibration isolator had a very high recovery, and the significantly deformed (12–20%) structure was restored to its original state without plastic deformation. The SMA-based vibration isolating element had a peak acceleration level and high vibration energy within a low-frequency band ($\leq$100 Hz), whereas the energy level in the high-frequency range over the critical frequency

was relatively reduced owing to "frequency decoupling". In addition, the structural deformation caused by the vibrating modes was restored to its original shape owing to its super elasticity, which was an inherent characteristic of the SMA, and the vibration responses were nearly impervious to the high-frequency modes owing to the high damping characteristics of the multilayered lamination. As there were two types of vibration-isolating elements in the lateral and axial directions, the SMA-based vibration isolator used in this study significantly reduced the vibration transmitted to the satellite from the launch vehicle. Section 4 presents the vibration reduction performance of the vibration isolator according to the acoustic/vibration load inside the fairing using environmental tests.

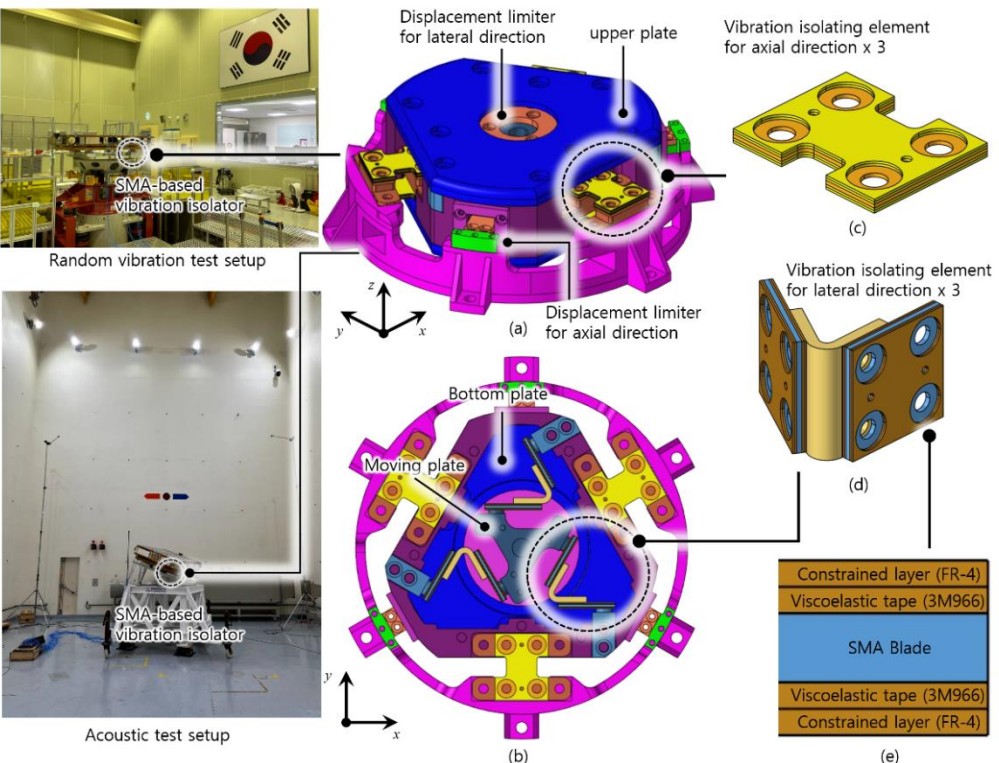

**Figure 2.** Configuration of SMA-based vibration absorber: (**a**) isometric view; (**b**) inside view; vibration isolating elements for (**c**) axial and (**d**) lateral directions with (**e**) lamination pattern in combination with a SMA blade, and viscoelastic layer [16].

## 4. Environment Test of the S-STEP Satellite under Random Vibration and Acoustic Loading

In this section, the vibration responses of the S-STEP satellite to random vibrations from the launch vehicle and the acoustic load generated in the fairing were investigated. Random vibration and acoustic tests were conducted with and without a vibration isolator. Random vibration and acoustic tests were conducted using the STM of the S-STEP satellite.

### 4.1. Introduction to the STM of the S-STEP Satellite

In this subsection, the STM of the S-STEP satellite, which was the target system for acoustic/random tests conducted in this study, is described. STM refers to the model used to experimentally verify the structural and thermal performance for achieving structural/thermal stability in an on-orbit/launch environment. Figure 3 shows a conceptual diagram of the S-STEP satellite. As aforementioned in Section 2, the S-STEP satellite had a plate-like structure in combination with an aluminum-skinned honeycomb panel, solar panel, SAR antenna, and another honeycomb with payload and electronics [3]. There was a cutout in the honeycomb structure in order to enhance mass reduction, except for the interface area attached to the electronics and payload structure. Thus, this composite structure provided an interface for the internal systems (PSU, battery, and IAU), and the

payload and bus structures shared a platform for installation. As this type of satellite had no deployment mechanism, the S-STEP satellite guaranteed high structural stability of the SAR antenna and thermal reliability of the internal components installed inside the S-STEP satellite.

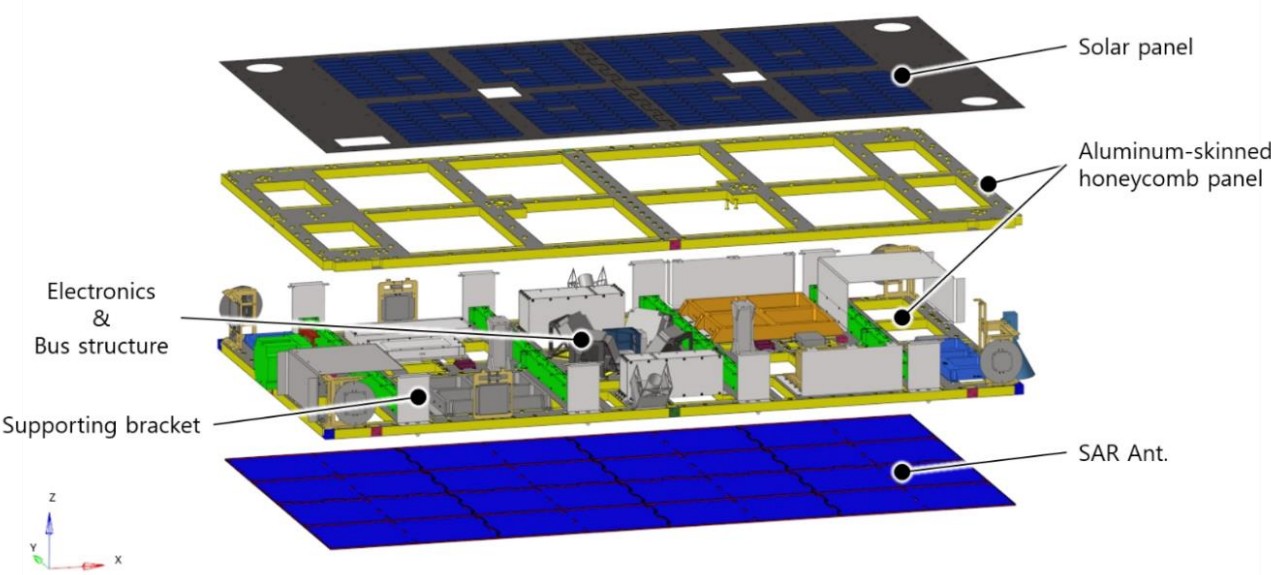

**Figure 3.** Conceptual diagram of the S-STEP satellite.

Figure 4 shows a section view of the S-STEP satellite with a vibration isolation system and launch adapter (24″ ring-type structure in Space X) [8]. The vibration isolation system consisted of an aluminum-skinned honeycomb panel, SMA-based vibration isolator (4EA), and in-orbit separation device (4EA). This system also had a flange bolt connecting the S-STEP satellite to the vibration isolator entirely isolated from the launch adaptor, using only the passive method based on the spring-damper system. The first natural frequency of the S-STEP satellite equipped with the vibration isolation system was set as 25 Hz in order to avoid the effects of the cutoff frequency [16]. Therefore, it was possible to realize vibration-free conditions in which the acoustic and vibrational loads transmitted to the satellite were significantly reduced. In this study, the performance of the vibration isolation system applied to the STM of an S-STEP satellite was investigated using acoustic and random vibration tests.

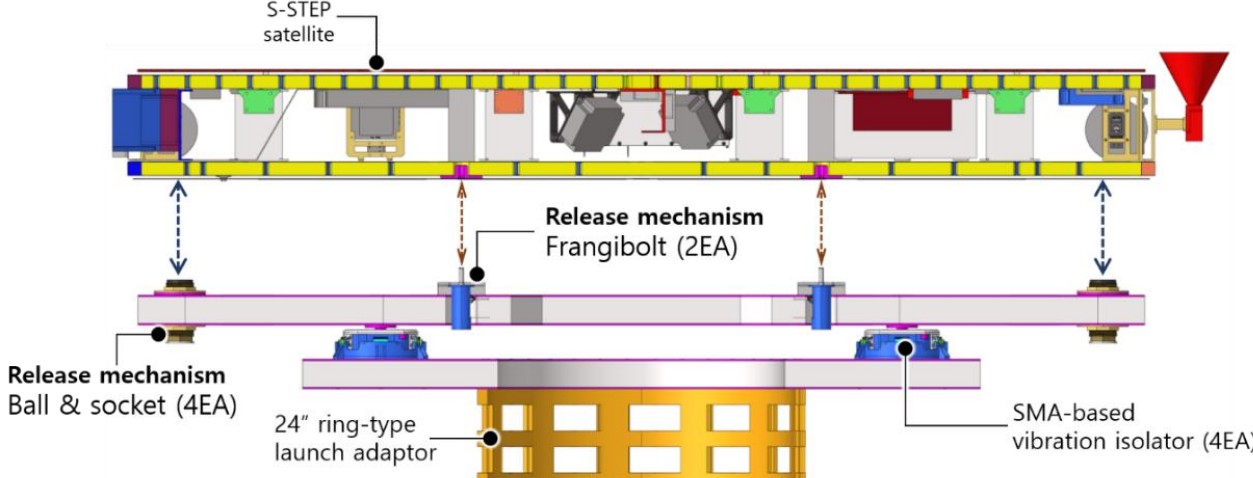

**Figure 4.** SMA-based vibration isolation system and STM of the S-STEP satellite.

### 4.2. Random Vibration Test of the S-STEP Satellite

Figure 5 shows the experimental setup of the random vibration test for the STM of the S-STEP satellite with SMA-based vibration. The qualification levels of the random vibration inputs transmitted from the launch vehicle are listed in Table 2. Using the setup in Figure 5 and the qualification level listed in Table 2, a random vibration test was performed to measure the acceleration response at the center of gravity (CoG), PSU, battery, and SAR antenna of the S-STEP satellite's STM.

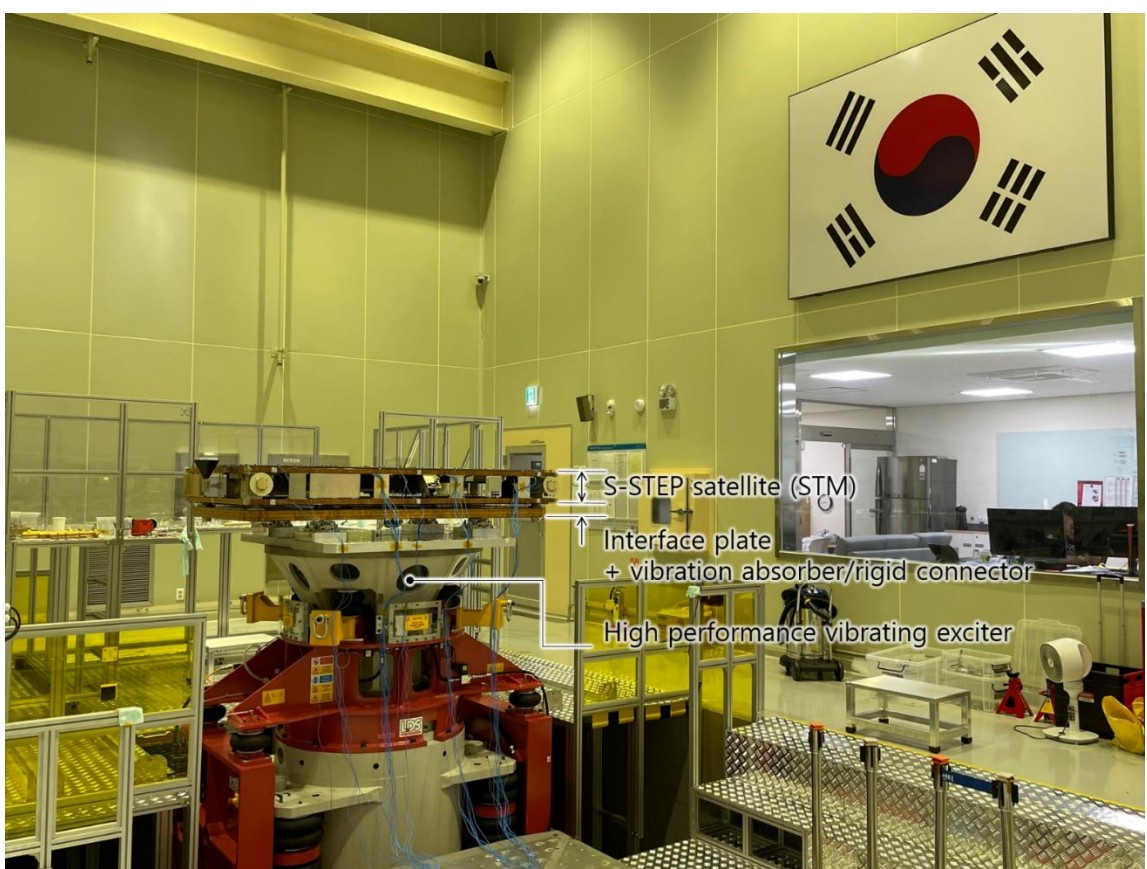

**Figure 5.** Experimental setup of a random vibration test for the STM of the S-STEP satellite.

**Table 2.** Qualification level of the vibrational profile for the random vibration analysis.

| Frequency [Hz] | Power Spectral Density (PSD) [g²/Hz] |
|:---:|:---:|
| 20 | 0.014 |
| 80 | 0.044 |
| 160 | 0.07 |
| 640 | 0.07 |
| 800 | 0.12 |
| 1150 | 0.12 |
| 1300 | 0.04 |
| 2000 | 0.04 |
| Root mean square (RMS) acceleration | 11.64 grms |

Figure 6 shows the design load of the random vibration test and frequency responses of the CoG, PSU, battery, and SAR antenna of the satellite. As shown in Figure 6a,b, the satellite with a vibration isolator showed an overall lower energy level than that of the rigid connector for the same random input. This was because the energy was concentrated in the low-frequency band (under 100 Hz) owing to the influence of the vibrating mode, and the overall vibration level (100–2000 Hz) was reduced by the high damping properties of the multilayered vibration-isolating elements. Figure 6c shows the root mean square (RMS)

acceleration of the S-STEP satellite with and without a vibration isolator. As shown in this figure, the overall vibrational response of the S-STEP satellite with the vibration isolator was lower than that of the rigid connector.

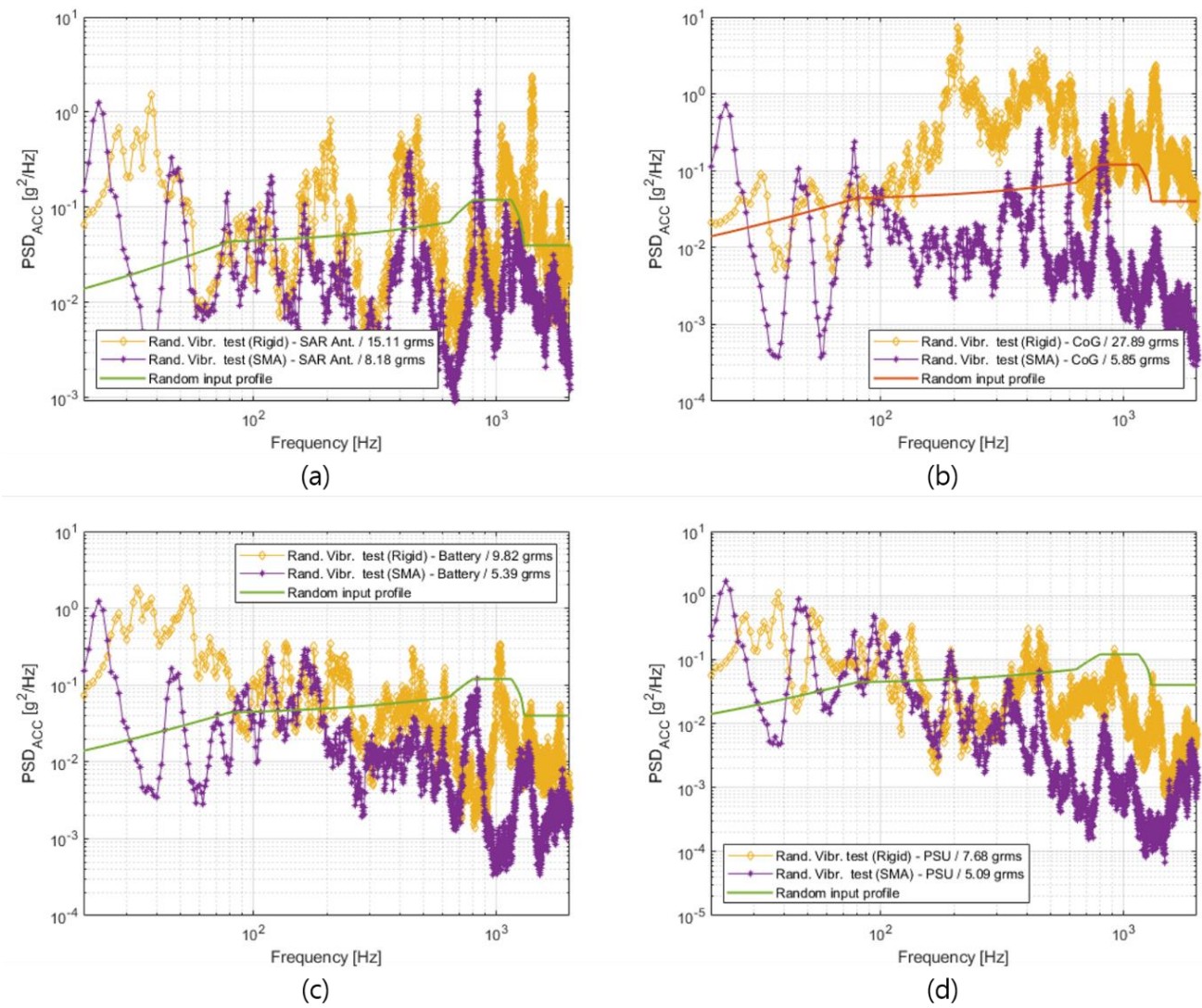

**Figure 6.** Random vibration test result for the STM of the S-STEP satellite at different locations with and without the SMA-based vibration isolator measured, and the random input profile: (**a**) SAR antenna; (**b**) CoG; (**c**) battery; (**d**) PSU.

### 4.3. Acoustic Test of the S-STEP Satellite in the Diffused Field

Figure 7 shows the experimental setup for the acoustic test of the STM of the S-STEP satellite. As mentioned before, the airflow and combustion/injection of fuel generates acoustic noises during the satellite launch. The acoustic noise in the launch environment was transmitted to the inside of the launch vehicle's fairing, and a diffuse sound field was generated in the fairing, which was an enclosed space. Owing to the characteristics of the diffused sound field, a significant acoustic energy was evenly formed in space in the fairing [23–25]. An acoustic test of the S-STEP satellite for the diffused sound field was conducted at the Korea Aerospace Research Institute (KARI). The volume of the acoustic chamber at KARI in Figure 7a was 1227 m³ ((W) 10.7 × (D) 8.5 × (H) 13.5 m), which was 800 times larger than that of the volume of the S-STEP satellite (1.5 m³). The test space was sufficient for the environment test of the diffused sound field inside the fairing. Figure 7b shows the S-STEP satellite installed in the testbed. For the acoustic test, the S-STEP satellite was mounted at an oblique angle of 25°. The horizontally installed plate-type

satellite focused on the acoustic energy between the acoustic source and satellite body. Consequently, the acoustic energy in the reverberation space was not uniformly distributed. In this way, the influence of the acoustic mode was avoided and a uniform diffused sound field was generated. Because the acoustic source was located on top of the chamber and the S-STEP satellite was a plate-type, the acoustic energy could be focused between the source and satellite, and the sound field in the chamber was no longer a diffused field. Figure 7c shows the acoustic source used to generate the diffused sound field. The target frequency band of the acoustic source was divided into low and high frequencies according to the shape and structure of the sources. The two sources produced sound through a cylinder-shaped double air valve reciprocated by pumping air. The acoustic profile was produced via feedback control using a microphone in the chamber.

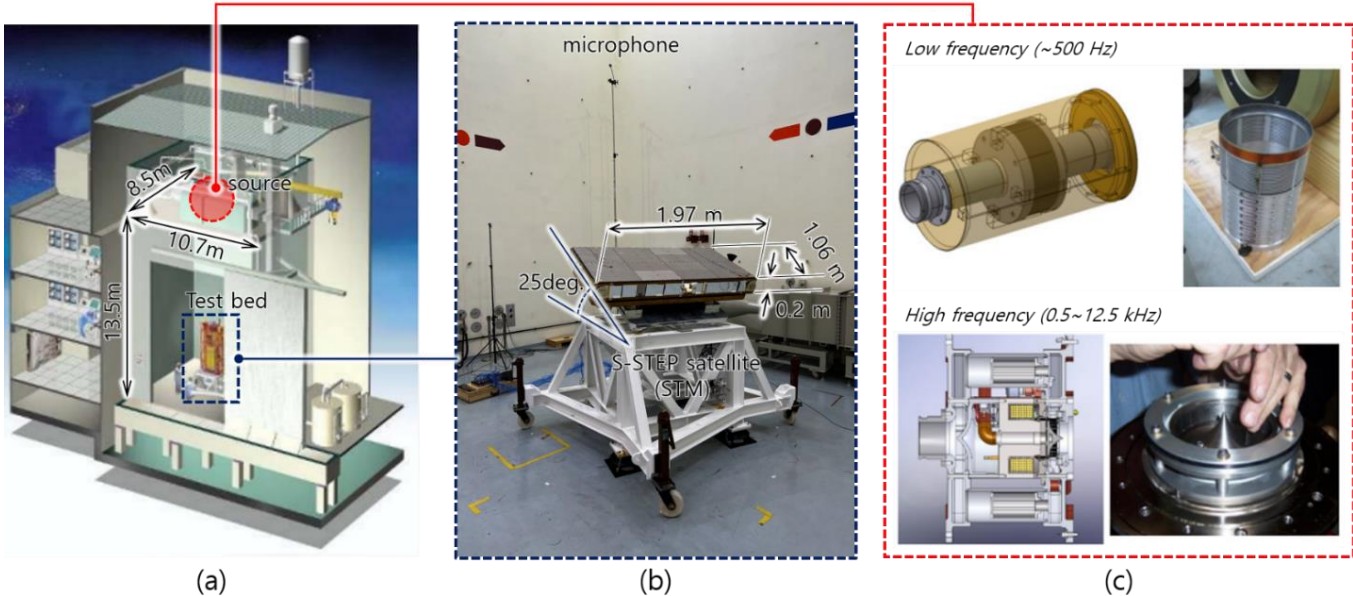

**Figure 7.** Experimental setup for the acoustic test of the S-STEP satellite (STM) subjected to the acoustic loading generated from the diffused field: (**a**) conceptual diagram of the reverberation chamber; (**b**) test bed for the S-STEP satellite (STM); (**c**) acoustic modulator to generate the sound field.

Figure 8 shows the acoustic input profile for the acoustic test performed in this study. As aforementioned, the satellite design can be determined by functional and environmental tests just before the launch process. In most cases, because the satellite design is decided after the certification process in which all functional and environmental tests are performed, the detailed design of the satellite can be changed during environmental testing. Moreover, it is difficult for a satellite designer to prepare an environmental test before the launch vehicle is decided. In this regard, the acoustic test input was determined as the envelope of the acoustic specifications of Vega-C, Falcon9, Soyuz, and Epsilon [8–11]. The acoustic environments of the several commercial launch vehicles were achieved with the test launch and realistic measurement of the acoustic pressure.

Figure 9 shows the design load of the random vibration test and frequency response of the CoG, PSU, battery, and SAR antenna. As shown in the acoustic test results in Figure 8a, the effect of the SMA-based vibration isolator on the acoustic load was relatively smaller than that of the random vibration test. Moreover, the RMS acceleration of the SAR antenna with the vibration isolator was higher than (0.5 grms) that of the result without the vibration isolator. However, the RMS acceleration of the satellite with the vibration isolator was smaller than that of the other cases, except for the SAR antenna. The SAR antenna of the satellite without the vibration isolator had a high RMS acceleration because the SAR antenna was directly excited by the diffused sound field.

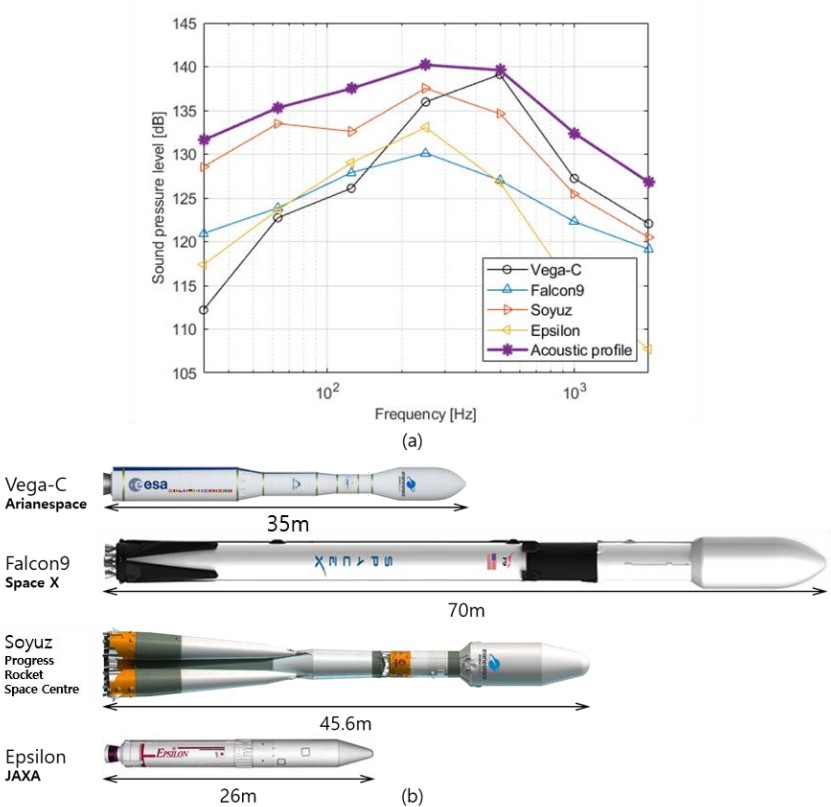

**Figure 8.** Specification of (**a**) the acoustic profile used in the acoustic test for the launch environment considering (**b**) several commercial launch vehicles of Vega-C, Falcon9, Soyuz, and Epsilon.

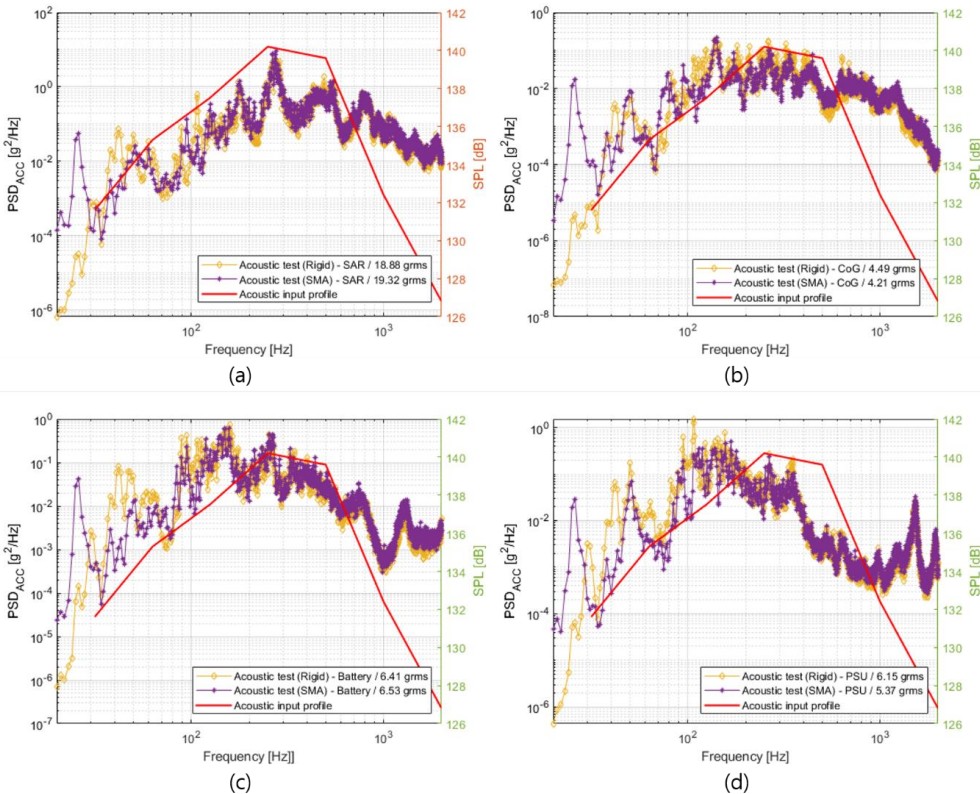

**Figure 9.** Acoustic test results for the STM of the S-STEP satellite subject to acoustic loads measured at different locations: (**a**) SAR antenna; (**b**) CoG (**c**) battery; (**d**) PSU with and without the SMA-based vibration isolator, and the acoustic input profile.

Figure 10 shows random vibration responses of the SAR antenna measured during the random vibration test for the SAR antenna level and acoustic test for the satellite level. As shown in Figure 10a, the specification level for the SAR antenna derived from the random vibration test was lower than that of the acoustic test result by a factor of 54%. However, the vibration level at the edge area was lower than that of the central area. In terms of energy, the random vibration result in the central area achieved from the SAR antenna level test enveloped the acoustic test result. The random vibration response did not cover the acoustic test result at partial frequency ranges (330–527 and 800~1350 Hz). Although the mechanical load of the acoustic test mainly excited the SAR antenna, the design load determined by the random vibration test was 100 g [3], whereas the maximum equivalent static loads at two partial frequency ranges were 77 and 85 g. These equivalent static loads, calculated by Mile's Equation [26] and ECSS rule [27], were relatively smaller than the design load. In addition, in this load condition, the SAR antenna assured the structural safety of the system. On the upper side of the satellite (not the bottom side), the results of the acoustic test showed the same trend for the random vibration test. This means that the random vibration test covered the acoustic test. Therefore, for cheaper, lighter, and faster development of spacecraft, the random vibration test can replace the acoustic test. As shown in Table 3.

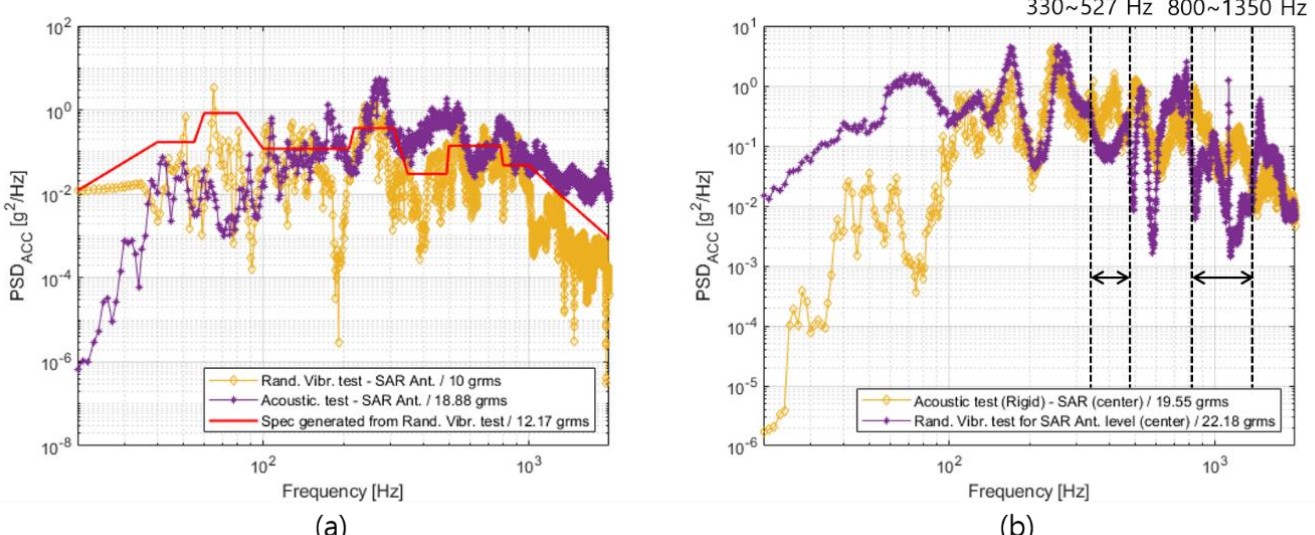

(a)    (b)

**Figure 10.** Random vibration responses of SAR antenna measurements from the random vibration test for the SAR antenna level and acoustic test for the satellite level: (**a**) edge area connected with satellite body; (**b**) central area of the satellite.

**Table 3.** Results of the low-level sine sweep (LLSS) test performed before and after the random vibration test.

| Part | Status | 1st Peak Frequency | Frequency Shift Difference (%) |
|---|---|---|---|
| SAR Ant. | Before | 26 | 3.8 |
|  | After | 25 |  |
| CoG | Before | 25 | 0 |
|  | After | 25 |  |
| Battery | Before | 25 | 0 |
|  | After | 25 |  |
| PSU | Before | 25 | 0 |
|  | After | 25 |  |

### 5. Conclusions

This study focused on an environmental test for an acoustic field and random vibration. During the launch, all of the satellites were subjected to severe mechanical loads of diffused sound field and random vibration. To reduce these mechanical loads, an SMA-based vibration isolator was applied at the interface between the launch adapter and satellite. Although the mechanical load of the acoustic test mainly excited the SAR antenna on the upper side (SAR antenna) of the satellite (not the bottom side (solar panel)), the results of the acoustic test showed the same trend as that of the random vibration test. Therefore, using the SMA-based vibration absorber, the environment test of the satellite can be simplified. In addition, the dynamic load from the acoustic and random vibration disturbance can be reduced and the vibration-sensitive component can have mechanical stability. From this perspective, SMA-based vibration isolators can contribute to saving the time and capital required for satellite development. These advantages have made it possible to develop satellites according to the new space paradigm, which is a trend in the space industry worldwide.

**Author Contributions:** Writing—original draft preparation, H.-G.K. and Y.-H.P.; writing—review and editing, H.-G.K.; data curation, H.-G.K. and S.-C.K.; supervision, K.-R.K. and Y.Y.; project administration, H.-U.O. and Y.S.; funding acquisition, S.-C.S. and Y.Y. All authors have read and agreed to the published version of the manuscript.

**Funding:** This research was supported by the Challengeable Future Defense Technology Research and Development Program (912777601-9127776-04) of Agency for Defense Development (ADD) in 2022.

**Data Availability Statement:** Not applicable.

**Conflicts of Interest:** The authors declare no conflict of interest.

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
