# Peer review of "Performance Investigation of Superplastic Shape Memory Alloy-Based Vibration Isolator for X-Band Active Small SAR Satellite of S-STEP under Acoustic and Random Vibration Environments"

_aerospace, doi:10.3390/aerospace9110642_

Round 1
Reviewer 1 Report
Dear Authors,
please check the correction in the attached .pdf file. All the parts that need review are highlighted in yellow. By clicking on it you will see the comment and/or request.

Author Response
First of all, thanks for your considerable review. Through the effort of you, our manuscript has a detail information and degree of completion. We revise our manuscript reflecting your opinion.
- I suggest to cancel the part in yellow because the launch cost is not necessarily smaller because the sum of the mass of the smaller satellites can be comparable to the mass of a single larger satellite. Unless you specify better the advantage for the launch process of a constellation of smaller satellites
“and launch process, and multiple satellites can be launched simultaneously, thereby significantly reducing the launch cost”
- Reflecting your opinion, we delete this phrase. - This sentence is not clear, what do you mean by mainteining the space environment, the space environment is what it is.
“In addition, considering the characteristics of the space environment, which is very difficult to maintain and has a limited power budget for the satellite system, passive vibration control is more suitable for reducing the vibration of the satellite system.”
- Reflecting your opinion, we inserted the additional information. - This is a repetition and so it must be deleted and the rest of the sentence has to be checked because it seems to me that the descriptions does not always corresponds to the numbers reported
“As seen in Figures 1(b) and (c), the S-STEP satellite was a compact plate-type structure weighing 80 kg, and the range, resolution, acquisition time, transmitted peak power, and polarimetry were 1 m, 5 km, 60 s, and 1 m, 5 km, 2560 W and, 60 s, respectively.”
- we edited this expression. this is grammar error. - Move this sentence after "different locations" in the line above. This because the way it is written now it seems that the sentence in yellow refers only to the PSU.
“Figure 5. Random vibration test result for the STM of the S-STEP satellite at different locations: (a) SAR antenna; (b) CoG; (c) battery; (d) PSU with and without the SMA-based vibration isolator measured, and random input profile.”
- We edited this figure caption1 - ADD: "because the horizontally ....not uniformly distributed" AND ADD ALSO "This way the influence of th acoustic mode is avoided and a uniform diffused sound field is generated"
“As shown in Figures 6(a) and (b), the acoustic source was located on top of the chamber, and the S-STEP satellite was a plate- type. The horizontally installed plate-type satellite focused on the acoustic energy between the acoustic source and satellite body. Consequently, the acoustic energy in the reverberation space was not uniformly distributed. Therefore, to avoid the influence of the acoustic mode and generate a uniform diffused sound field, the S-STEP satellite was installed at an oblique angle in the test bed”
- We edited this paragraph according to the logical flow. - Why you say that?
“As shown in Figures 6(a) and (b), the acoustic source was located on top of the chamber, and the S-STEP satellite was a plate-type.”
- If the satellite is parallel with the chamber floor, the acoustic energy focused between the source and satellite. This is no longer the diffused field. In order to make up the diffused sound field, representing EV condition in the fairing of the launch vehicle, we placed the satellite with the oblique angle. For the clear description, we edited this sentence. - This sencente does not seem a consequence of what written previusly and indeed does not add any useful information. I suggest to delete
Thus, before the launch process, it is difficult for a satellite designer to predict the types of vehicles that launch satellites.
- We deleted this sentence - SUBSTITUTE WITH "before the launch vehicle is decided.
"to predict which launch vehicle will be used and the launch environment.”
- We edited this expression according to your opinion. - This sencente does not seem a consequence of what written previusly and indeed does not add any useful information. I suggest to delete.
Thus, before the launch process, it is difficult for a satellite designer to predict the types of vehicles that launch satellites.
- We are deleted this sentence according to your opinion. - VEGA C is 35 meters. You may have confused the height of VEGA C with that of VEGA launcher that is 30 m. But now is VEGA C is in operation after its succesful inaugural flight of past 13 July 2022.
- We edited the size information of Vega-C and we also correct the manufacturer of Vega-C. - 1) SUBSTITUTE WITH "is basically negligible in all cases showing a small decrease but also a small increase in the case of the SAR antenna and of the battery"
was relatively smaller than that of the random vibration test.
2) DELETE FROM "Moreover up to SAR antenna"
Moreover, the RMS acceleration of the SAR antenna with the vibration isolator was higher than (0.5 grms) that of the result without the vibration isolator. However, the RMS acceleration of the satellite with the vibration isolator was smaller than that of the other cases, except for the SAR antenna.
- We edited above sentences according to your opinion. - This figure should be on the left and the figure 8a placed here. there is an exchange in the position of those two figures
- Reflecting on your opinion, we modified the figure 8 corresponding to the caption and body paragraph. - All this part is not clear at all. It has to be rewritten clearly. The conclusion is disputable. In this case it may be ok, but that cannot be generalized for any satellite or satellite part. Where the number 100g, 777g and 43g comes from? They were never mentioned before. Also figure 9 a and b are not clear. An effort should be made for improving readability and the understanding of the reader.
As shown in Figure 9(a), the specification level for the SAR antenna derived from the random vibration test was lower than that of the acoustic test result by a factor of 54 %. However, the vibration level at the edge area was lower than that of the central area. In terms of energy, random vibration result in the central area achieved from the SAR antenna level test enveloped the acoustic test result. The random vibration response did not cover the acoustic test result at partial frequency ranges (330–527 and 800~1350 Hz). The design load determined by the random vibration test was 100 g, whereas the maximum equivalent static loads at two partial frequency ranges were 777 and 43 g. In this load condition, the SAR antenna assured the structural safety of the system. Although the mechanical load of the acoustic test mainly excited the SAR antenna on the upper side of the satellite (not the bottom side), the results of the acoustic test showed the same trend for the random vibration test. It meant that the random vibration test covered the acoustic test. Therefore, for cheaper, lighter, and faster development of spacecraft, the random vibration test can replace the acoustic test.
- By self-reviewing the end of manuscript, this manuscript has some type off and logical errors. The number of 777 and 43 are type-off and the additional description of Figure 9. The design load of 100g was calculated from the random vibration test. - This acronym is reported only here you should specify its meaning
Table 2. Results of the LLSS test performed before and after the random vibration test.
- LLSS test mean low-level sine sweep test. We inserted the description of LLSS. - We also correct several grammar errors, reflecting your guide.
Details are shown in the attached file

Reviewer 2 Report
- The overall work in this manuscript likes a test report which lacks of methodology. The authors should provide the design principle of vibration isolation.
- It is well know that high damping material will reduce vibrations. The authors should explain more clearly the reason to choose the materials in Fig.2e.
- In addition, it is also necessary to provide the geometric parameter of each layer for Fig.2e.
- in Fig.7,please check the manufacturer of Soyuz rocket.
Author Response
First of all, thanks for your considerable review. Through the effort of you, our manuscript has a detail information and degree of completion. We revise our manuscript reflecting your opinion.
- The overall work in this manuscript likes a test report which lacks of methodology. The authors should provide the design principle of vibration isolation.
- This sentence was modified - 1) It is well know that high damping material will reduce vibrations. The authors should explain more clearly the reason to choose the materials in Fig.2e.
2) In addition, it is also necessary to provide the geometric parameter of each layer for Fig.2e.
- The design of SMA-based vibration isolator is out of scope. Geometrical parameters and material properties are shown in the following journal paper;
H. Park, S.C. Kwon, K.R. Koo, H.U. Oh, High damping passive launch vibration isolation system using superelastic SMA with multilayered viscous lamina. Aerospace 8.8 (2021): 201.
This paper was mentioned in the reference of this paper. Also, this paper contains the basic concept of the SMA-based vibration isolator. Two main working principles are the hyper elasticity of SMA material and high damping property of viscoelastic tape. This description was inserted into the manuscript. - In Fig.7,please check the manufacturer of Soyuz rocket.
- Maybe, this is type off. We correct this wrong information in Figure 7(b).
Details are shown in the attached file.

Reviewer 3 Report
General remarks
The main idea behind the paper is interesting: using a shape memory alloy as vibration isolator in order to reduce the random vibrations and acoustic vibrations of a satellite in the launch environment. The shape memory alloy-based vibration isolator used in this study reduces the vibration transmitted to the satellite from the launch vehicle. Two types of vibration-isolating elements are used: on the lateral direction and axial direction.
The title and the intentions declared in the abstract correspond to the contents of the paper. The paper contains an abstract and an introduction which is in fact a critical review of the state of the art. The authors have important contributions in this field in the last fifth years:
1. Park, T.Y.; Oh, H.U. Validation of the Critical Strain-Based Methodology for Evaluating the Mechanical Safety of Ball Grid Array Solder Joints in a Launch Random Vibration Environment. Journal of Electronic Packaging, Transactions of the ASME 2022, 144.
2. Kim, S.; Song, C.M.; Lee, S.H.; Song, S.C.; Oh, H.U. Design and Performance of X-Band SAR Payload for 80 kg Class Flat-Panel-Type Microsatellite Based on Active Phased Array Antenna. Aerospace 2022, 9.
3. Son, J.; Hongju, L.; Koo, K.R.; Park, J.H.; Kim, S.; Song, S.C.; Kim, H.; Oh, H.U. Mission and System Design of 80kg-class X-band Active SAR Satellite of S-STEP. 2021, B4.
4. Park, Y.H.; Park, J.H.; Park, S.W.; Kang, S.J.; Oh, H.U. Passive Vibration Suppression of Solar Array by Using Hyperelastic Shape Memory Alloy. 2021, C2.
5. Park, Y.H.; Kwon, S.C.; Koo, K.R.; Oh, H.U. High damping passive launch vibration isolation system using superelastic sma with multilayered viscous lamina. Aerospace 2021, 8.
6. Park, T.Y.; Kim, S.Y.; Yi, D.W.; Jung, H.Y.; Lee, J.E.; Yun, J.H.; Oh, H.U. Thermal Design and Analysis of Unfurlable CFRP Skin-Based Parabolic Reflector for Spaceborne SAR Antenna. International Journal of Aeronautical and Space Sciences 2021, 22, 433-444.
7. Park, T.Y.; Chae, B.G.; Kim, H.; Koo, K.R.; Song, S.C.; Oh, H.U. New thermal design strategy to achieve an 80-kg-class lightweight x-band active sar small satellite s-step. Aerospace 2021, 8.
8. Noh, H.K.; Lim, J.H.; Kwon, S.C.; Jeong, S.K.; Park, T.Y.; Oh, H.U. A comparative study of fatigue life prediction methodologies for electronic PCB under random vibration. Transactions of the Korean Society of Mechanical Engineers, A 2021, 45, 905-913.
9. Kwon, S.C.; Son, J.H.; Song, S.C.; Park, J.H.; Koo, K.R.; Oh, H.U. Innovative mechanical design strategy for actualizing 80 kg-class x-band active sar small satellite of s-step. Aerospace 2021, 8.
10. Kim, H.I.; Son, M.Y.; Oh, H.U. Thermal Design of Cryogenic Cooler by Using Graphite Sheet for Enhancement of Micro-vibration Isolation Performance. 2021, C2.
11. Go, J.S.; Choi, J.S.; Park, S.W.; Kang, S.J.; Oh, H.U. Experimental Validation of Deployable Tape Spring Hinge Combined with Surperelastic Shape Memory Alloy. 2021, C2.
12. Park, Y.H.; Jo, M.S.; Lee, E.S.; Oh, H.U. Performance Enhancement of Spaceborne Cooler Passive Launch and On-Orbit Vibration Isolation System. International Journal of Aerospace Engineering 2020, 2020.
13. Park, T.Y.; Oh, H.U. Structural design methodology of spaceborne electronics for implementing lightweight and low-cost small satellite applications. 2020, 2020-October.
14. Kwon, S.C.; Jo, M.S.; Ko, D.H.; Oh, H.U. Viscoelastic multilayered blade-type passive vibration isolation system for a spaceborne cryogenic cooler. Cryogenics 2020, 105.
15. Kwon, S.C.; Oh, H.U. Passive micro-jitter isolation of gimbal-type antenna by using a superelastic SMA gear wheel. Mechanical Systems and Signal Processing 2019, 114, 35-53.
16. Kim, T.; Oh, H.U. Experimental Investigation of the Feasibility of Using a Liquid Metal as a Variable Conductance Radiator for Space Applications. Journal of Heat Transfer 2019, 141.
17. Park, T.Y.; Park, J.C.; Oh, H.U. Evaluation of structural design methodologies for predicting mechanical reliability of solder joint of BGA and TSSOP under launch random vibration excitation. International Journal of Fatigue 2018, 114, 206-216.
18. Park, T.Y.; Kim, S.H.; Kim, H.; Oh, H.U. Experimental Investigation on the Feasibility of Using Spring-Loaded Pogo Pin as a Holding and Release Mechanism for CubeSat's Deployable Solar Panels. International Journal of Aerospace Engineering 2018, 2018.
19. Kwon, S.C.; Onoda, J.; Oh, H.U. Improvement of micro-jitter energy harvesting efficiency of piezoelectric-based surge-inducing optimal switching strategy. Sensors and Actuators, A: Physical 2018, 281, 55-66.
20. Park, T.Y.; Park, G.J.; Oh, H.U. Enhancement of thermal control performance by using liquid metal radiator. 2017, 12, 8117-8122.
21. Oh, H.U.; Lee, M.J.; Kim, T. Experimental Design Validation of Tilting Calibration Mechanism by Using Shape Memory Alloy Spring Actuator. International Journal of Aerospace Engineering 2017, 2017.
22. Kwon, S.C.; Park, Y.H.; Oh, H.U. Characteristics of spaceborne cooler vibration isolator using a pseudoelastic shape memory alloy. 2017, 10168.
23. Kwon, S.C.; Jo, M.S.; Oh, H.U. Experimental validation of fly-wheel passive launch and on-orbit vibration isolation system by using a superelastic SMA mesh washer isolator. International Journal of Aerospace Engineering 2017, 2017.
24. Kwon, S.C.; Jeon, Y.H.; Oh, H.U. Micro-jitter attenuation of spaceborne cooler by using a blade-type hyperelastic shape memory alloy passive isolator. Cryogenics 2017, 87, 35-48.
Some remarks and questions
1. The affiliation of the last author is missing (Hyun-Ung Oh).
2. I consider that the acronyms should not be defined in the abstract; the acronyms should be defined in other sections (i.e., Introduction, Material and methods, Results, and Discussion). Likewise, the acronyms must be defined only once and when they are first presented.
3. I recommend introducing the shape memory alloy (SMA) as keyword.
4. The state of the art can be more detailed. This chapter contains 25 references, 8 references of the authors of the present paper. Remember that all references must be in accordance with the main topic of the research work, methods, results, and discussions; avoid alteration of citations leading to bad scientific practices favouring authors or journals.
5. It is usual to put at the end of the Introduction (State of the art) chapter some paragraphs that identify what novelty the paper brings with respect to other papers. Can the authors emphasize this?
6. The paper does not contain a Materials and Methods chapter where the authors should present the experimental layout used in the experiments. Also it is important to present the chemical composition and the properties of the shape memory alloy.
7. Please explain what PSD represents in Table 2. I think, Power Spectral Density, but I consider that is necessary to put it on the paper.
8. Please rephrase this: “This doubled the transmitted vibration from the launch vehicle and excited the satellite.”
9. The Figure 2 was also presented in the paper: Park, Y.H.; Kwon, S.C.; Koo, K.R.; Oh, H.U. High damping passive launch vibration isolation system using superelastic SMA with multilayered viscous lamina. Aerospace 2021, 8. I recommend changing it bringing some new ideas in the figure.
10. Can the authors elaborate on the following phrase: “the acoustic test input was determined as the envelope of the acoustic specifications of Vega-C, Falcon9, Soyuz, and Epsilon”. How did they create a generic environment for the acoustic test?
11. Please correct the phrase: “As shown in the acoustic test results in Figure 7” with “As shown in the acoustic test results in Figure 7(a)” – page 11, lines 292-293.
12. Take care about the format of the reference section. The authors put the number of some references twice.
13. I request that the authors report conclusions coherent, concise, and concrete according to the problem statement and the main objectives of the research work.
The paper may be accepted considering the responses received for the above queries raised.
Author Response
First of all, thanks for your considerable review. Through the effort of you, our manuscript has a detail information and degree of completion. We revise our manuscript reflecting your opinion.
- The affiliation of the last author is missing (Hyun-Ung Oh).
- The last author in corresponding author. We insert the affiliation information of the last author and modified 1st - I consider that the acronyms should not be defined in the abstract; the acronyms should be defined in other sections (i.e., Introduction, Material and methods, Results, and Discussion). Likewise, the acronyms must be defined only once and when they are first presented.
- Reflecting your opinions, all the acronyms are removed from the original manuscript. - I recommend introducing the shape memory alloy (SMA) as keyword.
- Regarding with Question #2, acronyms of S-STEP, STM are also inserted to keywords section. - The state of the art can be more detailed. This chapter contains 25 references, 8 references of the authors of the present paper. Remember that all references must be in accordance with the main topic of the research work, methods, results, and discussions; avoid alteration of citations leading to bad scientific practices favouring authors or journals.
- I entirely agree with your opinion. That is very bad scientific customs. We need an effort to avoid this kind of self-citation. Because this paper was written by the view point of the system engineering for the S-STEP satellite, it is necessary and inevitable to describe the historical review of small SAR satellite. But, this paper is about the environment test. So, this paper mainly describes the EV spec and test setup. All the literatures cited by this manuscript are essential. We promise that several other papers are cited in the next our paper. - It is usual to put at the end of the Introduction (State of the art) chapter some paragraphs that identify what novelty the paper brings with respect to other papers. Can the authors emphasize this?
- Yes, we can. There are two main contributions in this paper. First is to experimentally identify the vibration reduction performance of the SMA-based vibration isolator. Second is to compare the vibrational responses measured by the acoustic and random vibration test. In order to highlight this point, we modified the manuscript. - The paper does not contain a Materials and Methods chapter where the authors should present the experimental layout used in the experiments. Also it is important to present the chemical composition and the properties of the shape memory alloy.
- Although, there is no the section titled with “Materials” or “Methods”, the information for describing the research plan and result was sufficiently shown in the manuscript. The experimental layout is shown in Figure 4 and 6. Details about material properties and geometrical parameters are shown in the following literature;
H. Park, S.C. Kwon, K.R. Koo, H.U. Oh, High damping passive launch vibration isolation system using superelastic SMA with multilayered viscous lamina. Aerospace 8.8 (2021): 201.
Design philosophy of the SMA-based vibration isolator is shown in above literature. Please note that this paper focus on the experimental verification for the acoustic and random vibration tests. - Please explain what PSD represents in Table 2. I think, Power Spectral Density, but I consider that is necessary to put it on the paper.
- We insert the full words of PSD into the table 2 - Please rephrase this: “This doubled the transmitted vibration from the launch vehicle and excited the satellite.”
- This sentence was rephrased as follow; - The Figure 2 was also presented in the paper: Park, Y.H.; Kwon, S.C.; Koo, K.R.; Oh, H.U. High damping passive launch vibration isolation system using superelastic SMA with multilayered viscous lamina. Aerospace 2021, 8. I recommend changing it bringing some new ideas in the figure.
- As mentioned before, this paper focus on the experimental verification of the SMA-based vibration isolator shown in the literature (High damping passive launch vibration isolation system~). So, the overall layout and material properties are same with the previous research. Considering the research objective, we modified Figure 2. - Can the authors elaborate on the following phrase: “the acoustic test input was determined as the envelope of the acoustic specifications of Vega-C, Falcon9, Soyuz, and Epsilon”. How did they create a generic environment for the acoustic test?
- These acoustic profiles were achieved with the test launch and realistic measurement of the acoustic pressure. Reflecting your opinion, the description was inserted. - Please correct the phrase: “As shown in the acoustic test results in Figure 7” with “As shown in the acoustic test results in Figure 7(a)” – page 11, lines 292-293.
- Reflecting your opinion, we modified the phrase - Take care about the format of the reference section. The authors put the number of some references twice.
- We reformatted the references. - I request that the authors report conclusions coherent, concise, and concrete according to the problem statement and the main objectives of the research work.
- Considering concision, concreteness, and consistency, we improve the conclusions section.
Details are shown in the attached file.
